# Heated Tobacco Products and Nicotine Pouches: A Survey of People with Experience of Smoking and/or Vaping in the UK

**DOI:** 10.3390/ijerph18168852

**Published:** 2021-08-22

**Authors:** Leonie S. Brose, Máirtín S. McDermott, Ann McNeill

**Affiliations:** 1Addictions, Institute of Psychiatry, Psychology and Neuroscience, King’s College London, 4 Windsor Walk, London SE5 8BB, UK; mairtin.mcdermott@gmail.com (M.S.M.); ann.mcneill@kcl.ac.uk (A.M.); 2SPECTRUM Consortium, Edinburgh EH8 9YL, UK

**Keywords:** nicotine, novel tobacco products, smoking, inequalities

## Abstract

*Background*: To gauge the public health impact of new nicotine products, information is needed on use among different populations. Aims were to assess in adults who smoked, vaped, did both or had recently stopped: (1) awareness, ever and current use of heated tobacco products (HTPs) and nicotine pouches (NP), (2) characteristics associated with ever use, (3) reasons for use of and satisfaction with HTPs, (4) characteristics associated with interest in use of HTPs. *Methods*: Online survey in the UK in 2019, *n* = 3883. (1) Proportion aware, ever and current (≥monthly) use; (2) ever use regressed onto socio-demographics and smoking/vaping; (3) frequency of reasons for HTP use and satisfaction; (4) interest in trying HTPs regressed onto socio-demographics and smoking/vaping status. *Results*: Awareness was 34.8% for HTP and 15.9% for NP; current use was 3.2% and 2.7%. Being <45 years, higher education, living in London and currently both smoking and vaping were associated with ever having used the products. Curiosity was the most common reason for HTP use (79.8%) and 72.0% of ever HTP users found them at least as satisfying as smoking. Among those not currently using HTPs, 48.5% expressed any interest—lower among those aged over 65 and higher among those smoking and vaping. *Conclusions*: In this sample of adults with a history of nicotine use, very few currently used heated tobacco products or nicotine pouches. Satisfaction with and interest in HTPs were substantial. The low level of use is unlikely to substantially reduce the public health impact of smoking.

## 1. Introduction

Cigarette smoking remains the most common way of consuming nicotine [1] and the second most important risk factor for death and loss of healthy life years globally [2,3], almost entirely due to the effects of combustion, not nicotine [4]. Harm from smoking is concentrated among groups with other disadvantages such as low education or poor physical or mental health [5,6,7] and thus increases inequities.

New categories of nicotine-containing products have emerged in recent years, including heated tobacco products (HTPs) and nicotine pouches. HTPs are electronic devices that heat processed tobacco with the aim of avoiding combustion [8]. There are different types of HTPs, each produced by one of the major tobacco companies: IQOS (Philip Morris International), the first HTP to become available in the UK from late 2016 [8], and glo (British American Tobacco) both heat tobacco sticks to produce an aerosol that the user can inhale; iFuse (BAT) and Ploom Tech (Japan Tobacco International) heat an e-liquid to produce an aerosol which then passes over the tobacco. Philip Morris International had requested a risk modification and an exposure modification order for IQOS in the US. The US Food and Drug Administration (FDA) has authorised IQOS to be marketed as a modified risk tobacco product [9]. The FDA concluded that, based on current evidence on exposure in the absence of long-term studies, switching completely from smoking to IQOS reduced exposure to harmful or potentially harmful chemicals. However, they did not approve that IQOS use reduces the risk of tobacco-related diseases or reduces harm relative to continued cigarette smoking. Notably, in Japan, the introduction and rapid increase in use of HTP [10,11] has led to an acceleration in the decline of cigarette sales [12,13].

Nicotine pouches entered the UK market in the period 2018–2019. In appearance, they resemble snus, a type of tobacco pouch widely used in Sweden with substantially reduced harm compared with smoking [14,15]. However, in contrast to snus, nicotine pouches do not contain tobacco but nicotine-containing powder [16]. They are designed to be placed into the mouth to release nicotine. Brands include ZYN (Swedish Match), LYFT (British American Tobacco) and Nordic Spirit (Japan Tobacco International). Very little research on these products has been published. One study compared pharmacokinetics of ZYN with two other oral products, snus (tobacco pouches) and wet snuff (chewing tobacco). Results indicated that ZYN could deliver nicotine into the bloodstream as quickly and to a similar extent as the existing smokeless products with no significant adverse effects [17]. For Zonnic, an older type of nicotine pouch, trials found that compared with nicotine gum, nicotine pouches were as effective or more effective at reducing craving and favoured by smokers [18,19].

The net public health effect from the introduction of alternative nicotine products depends on the degree to which use is less harmful than smoking, whether people who smoke switch to other products, whether people who used to smoke take up new products, whether people who would not have used any nicotine or tobacco product take up new products and whether this diverts them from or attracts them to using more harmful products [20,21]. Understanding factors associated with use of novel products has important implications for gauging their public health effects [22].

There is some evidence on prevalence of and characteristics associated with HTP (mostly IQOS) use from nationally representative surveys in some countries. In the US in 2017, 1.1% of adults reported current use [23]. In Great Britain in 2017, 0.8% [24] and in 2019, 0.6% of adults indicated any current use [25]. In Germany in the period 2016–2017, 0.3% of current and recent smokers reported current HTP use [26]. In Japan in 2018, 2.7% of adults reported at least monthly use [27]. In South Korea in 2018, 4.4% of adults reported current use [28]. Use was very rare [24,27] or non-existent [28] among those who had not smoked, higher among those with higher education or income [24,26,28] and higher among younger age groups [24,26,28], although this was not the case in Japan [27]. A survey in Australia, Canada, England and the US in 2018 among current and former smokers and/or vapers found overall just under one-third were aware of HTPs, 2.4% had ever tried and 0.9% were current at least monthly users, with trial and current use higher among those who also smoked and vaped (8.4%) [29].

Almost no information is available on the use of nicotine pouches. One US study surveyed 1266 users of ZYN. Among them, 43% were former tobacco users (either of smokeless tobacco, cigarettes, both cigarettes and smokeless tobacco, or former e-cigarette and other combustible users), 26% were current smokeless tobacco users, 19% dual users, 8% current smokers, and 4% of had not smoked at least 100 cigarettes or regularly used other tobacco products, although all were currently using one or more tobacco products [21]. An analysis of smokeless tobacco products (including tobacco-free nicotine pouches) sold in US convenience stores reported that nicotine pouches accounted for 4% of unit sales by 2019 [30].

### Aims

To assess awareness, ever and current use of HTPs and nicotine pouches in a UK sample of people who smoked, vaped, did both or had recently stopped.To assess respondent characteristics associated with ever use of HTPs and nicotine pouches.To assess reasons for use of and satisfaction with HTPs among ever and current HTP users.To assess interest in use of heated tobacco among those who had never used or tried them up to a few times and characteristics associated with interest.

## 2. Materials and Methods

This cross-sectional study used self-reported data from one wave of a UK longitudinal online survey of adults who were current or former smokers and/or vapers. Survey participants were recruited through an online consumer panel of Ipsos Interactive Services. Data were collected and managed by Ipsos and anonymised to the researchers. The survey imposed quotas for age, gender and geographical region to ensure that the sample was representative. The first wave of data collection was in 2012 and the sample has been replenished twice. In wave 6 in September/October 2019, 1000 continuing and 2883 new participants (3883 in total) completed the survey.

### 2.1. Measures

Socio-demographics included gender (male, female), age (18–24, 25–34, 35–44, 45–54, 55–65, over 65), education (no university, some university), ethnicity using 2011 UK census categories [31], and UK region (England: North East, North West, Yorkshire and The Humber, West Midlands, East Midlands, East of England, South West, South East, Greater London; Wales; Scotland; Northern Ireland).

Measures related to different nicotine or tobacco products are provided in [Table ijerph-18-08852-box001].
Definitions of the different products were provided at the beginning of the survey and were available for each question. In the survey, the term ‘Heat-not-burn tobacco products (also known as Heated Tobacco Products)’ was used. Smoking and vaping status were assessed and combined into one measure (smoking only, smoking and vaping, vaping only, currently not smoking or vaping).

**Box 1 ijerph-18-08852-box001:** Nicotine product questions.

Heard about HTPs
Before this survey, had you heard about new electronic products that heat tobacco instead of burning it? These are battery-powered devices that heat tobacco in the form of capsules, pods or cigarette-like sticks instead of burning it to generate an aerosol that is inhaled by the user. They differ from e-cigarettes in that they heat tobacco, rather than an e-liquid. These include products such as IQOS.YesNoDon’t know
**HTP ever use**Have you ever used one of these “heat-not-burn” products, even one time?YesNoDon’t know
**HTP seen for sale**Have you ever seen any of these heat-not-burn products for sale in a shop or online?YesNoDon’t know
**HTP use**How often, if at all, do you CURRENTLY use heat-not-burn products?DailyLess than daily, but at least once a weekLess than weekly, but at least once a monthLess than once a month, but occasionallyNo, but I have tried a heat-not-burn product a few times (more than once)No, but I have tried a heat-not-burn product onceDon’t know*Responses a to f: ever use, a to c: current use, d to f: tried, g: excluded*
**HTP brand**Which heat-not-burn products have you ever used?iQOS with HeatSticks /HeetsOther (specify)Don’t know
**Interest in HTP use**Would you be interested in trying one of these heat-not-burn products [again] if you had the opportunity?Very interestedSomewhat interestedA little interestedNot interested at allDon’t know*Responses a to c categorised as any interest.*
**Reasons for HTP use**Which of the following were reasons for your using heat-not-burn products?They may not be as bad for your health as smoking tobacco cigarettesI like the flavoursThey make it easier for you to cut down on the number of cigarettes you smokeSo you can use them in places where smoking tobacco or cigarettes is bannedThey might help you quit smokingThey are cheaper than cigarettesFriends or family use themThey smell better than cigarettesThey don’t produce ash or buttsNo tobacco smokeThey may be more socially acceptableThe technologyA health professional advised you to do soBecause I like the tasteBecause I enjoy itJust to give it a try/ I was curious about themOther
**Relative satisfaction**How satisfying is using a heat-not-burn product compared to smoking tobacco cigarettes?More satisfying than tobacco cigarettesEqually satisfyingLess satisfying than tobacco cigarettesDon’t know
**Heard about nicotine pouches**Before this survey, had you heard about new nicotine pouches? These are small white sachets that do not contain any tobacco which people put in their mouth. They are relatively new on the market.YesNoDon’t know
**Nicotine pouch ever use**Have you ever used one of these nicotine pouches, even one time?YesNoDon’t know
**Nicotine pouch seen for sale**Have you ever seen any of these nicotine pouch products for sale in a shop or online?YesNoDon’t know
**Nicotine pouch use**How often, if at all, do you CURRENTLY use nicotine pouches?DailyLess than daily, but at least once a weekLess than weekly, but at least once a monthLess than once a month, but occasionallyNo, but I have tried a nicotine pouch a few times (more than once)No, but I have tried a nicotine pouch product onceDon’t know*Responses a to f: ever use, a to c: current use, d to f: tried, g: not selected*
**Smoking status****^1^**In this question, we are referring only to tobacco cigarettes and other smoked tobacco products (not vaping devices, e-cigarettes or heat-not-burn tobacco products). Could you please tell us which of the following best applies to you now? I smoke tobacco cigarettes (including hand-rolled) every dayI smoke tobacco cigarettes (including hand-rolled), but not every dayI do not smoke tobacco cigarettes at all, but I do smoke tobacco of some kind (e.g., pipe, cigar or shisha)I stopped smoking tobacco products (cigarettes, pipes, cigars, shisha etc) completely within the last 12 monthsI stopped smoking tobacco products (cigarettes, pipes, cigars, shisha etc) completely over 1 year agoI have never been a smoker (of cigarettes, pipes, cigars shisha etc)
**Vaping status****^1^**Could you please tell us which of the following best applies to you now? I currently vape/ use e-cigarettes dailyI currently vape/ use e-cigarettes but not every dayI have tried vaping/ an e-cigarette once or a few timesI stopped vaping/ using e-cigarettes completely within the last 12 monthsI stopped vaping/ using e-cigarettes completely over 1 year agoI have never vaped/ used e-cigarettes

^1^ Combined smoking and vaping status: Smoking status a to c and vaping status c to f = Smoking only; Smoking status a to c and vaping status a to b = Smoking and vaping; Smoking status d to f and vaping status a to b: Vaping only; Smoking status d to f and vaping status c to f: Not currently smoking or vaping.

### 2.2. Sample

All 3883 respondents were included to address aims 1 and 2. To address aim 3, the 242 respondents who had ever used HTPs were included for reasons for use and the subset of 193 respondents who had used HTPs more than once to describe satisfaction. To address aim 4, the 1199 respondents who had not used HTPs or had tried them up to a few times only were included in analysis.

### 2.3. Analyses

To address aim 1, the prevalence of having heard of, ever use and at least monthly us (current use) of HTP and nicotine pouches was described.

To address aim 2, associations between ever use of HTPs and nicotine pouches and socio-demographics and smoking and vaping status were assessed using bivariate and multivariable logistic regressions. Multivariable logistic regressions included gender, age, education, ethnicity, region and smoking and vaping status. Only ever use was used in regressions because current use was low. The results section reports multivariable (adjusted) analysis unless otherwise reported.

To address aim 3, frequency of responses to a list of possible reasons for use were described and compared using chi-square statistics for current users of HTPs and for those who had only tried HTPs or were using it less than monthly. Satisfaction with HTPs compared with cigarettes was described for those two groups.

To address aim 4, proportions interested in trying HTPs were described; bivariate and multivariable logistic regressions were used to assess associations with socio-demographics and smoking and vaping status. Multivariable logistic regressions included gender, age, education, ethnicity, region and smoking and vaping status and adjusted results are reported unless otherwise reported. Information on reasons for use, satisfaction or interest in use was not available for nicotine pouches.

## 3. Results

The sample was split evenly between men and women and covered all age groups. The majority (90.9%) identified as white and 45.9% had started or completed university education. Over two-thirds (69.4%) were current smokers and 37.3% were current vapers (Table 1). Combining those two measures showed that almost half of the sample were smoking but not vaping (46.9%), almost a quarter were smoking and vaping (22.4%), with the rest split between those vaping but not smoking (14.8%) and those neither smoking or vaping (15.8%).

### 3.1. Awareness, Ever Use and Current Use of HTP and Nicotine Pouches

Over one-third (34.8%) had heard about HTPs and 15.9% had heard about nicotine pouches. Fewer had ever seen HTPs (10.1%) or nicotine pouches (3.1%) for sale in shops or online. Heated tobacco products had ever been used by 6.2% of the sample, including 3.2% current users. Nicotine pouches had ever been used by 4.4%, including 2.7% current users. Of those who had ever used HTPs, 32.6% had used them only once or a few times; for nicotine pouches, this was 29.8% of all ever users (Table 1).

### 3.2. Characteristics Associated with Ever Use of HTPs and Nicotine Pouches

Men and women were equally likely to have ever used HTPs. Age groups under 45 years were more likely to have ever used than the oldest age group. Respondents with some university education were more likely to have ever used HTPs, and those from London were more likely to have ever used HTPs than respondents from other parts of England with Wales, Scotland and Northern Ireland similar to England (excluding London). Respondents currently vaping and smoking were more likely to have ever used HTPs than those who were exclusively smoking; those currently not smoking or vaping and those exclusively vaping had similar levels of ever use as those exclusively smoking (Table 2). In unadjusted analysis (Appendix A), there was also an association with ethnicity, with respondents categorised as not white more likely to have ever used.

For ever use of nicotine pouches, there was a gender difference with men more likely to have ever used. The associations with age, ethnicity, education, region and smoking/vaping had a similar pattern to those for HTP ever use. Groups more likely to have ever used were those aged under 45, those with university education and those from London. Those currently vaping and smoking were more likely to have ever used nicotine pouches than those who were exclusively smoking; those currently not smoking or vaping and those exclusively vaping had similar levels of ever use as those exclusively smoking (Table 2). Additionally, as with HTPs, an association with ethnicity was only apparent in unadjusted analysis (Appendix A).

### 3.3. Reasons for Use of HTPs and Satisfaction

Among those who had ever tried HTPs, the most commonly endorsed reason was ‘Just to give it a try/I was curious’ (79.8%) and with the exception of ‘A health professional advised you to do so’ and ‘other’, all reasons were endorsed by more than half of ever users (Table 3). Current users were more likely to support each reason than those who had only tried once or a few times, with the exception of ‘Just to give it a try/I was curious’ where those who had tried were more likely to agree, and ‘No tobacco smoke’ and ‘They don’t produce ash or butts’ where differences were not significant (Table 3).

Overall, 72.0% of those who had used HTPs more than once rated HTPs as at least as satisfying as smoking (Table 3). Perceived satisfaction relative to smoking was higher among those currently using HTPs (87.1% as least as satisfying) than among those who had tried a few times or were using HTPs less than monthly (44.9%, Table 3).

### 3.4. Interest in Use of HTPs

Among those who had never used HTPs or tried once or a few times, 48.5% reported any interest in trying HTPs (10.5% very interested, 17.8% somewhat interested, 20.2% a little interested), whereas 47.0% were not at all interested and 4.6% did not know. All age groups up to the age of 65 were more likely to report any interest in trying HTPs than those aged over 65. Those not smoking or vaping were less likely to report any interest and those currently vaping and smoking more likely to report any interest than those currently exclusively smoking (Table 2). In unadjusted analysis, those with some university education were also more likely to report any interest (Appendix A).

## 4. Discussion

In this sample comprising smokers, recent ex-smokers and vapers, a minority were aware of new nicotine products, with very small minorities having experience of using them. Approximately one-third were aware of HTPs, with ever and current use being just over six and three per cent, respectively. Corresponding figures for nicotine pouches were just over one in six being aware and ever and current use being just over four and just below three per cent, respectively. Being under 45 years, having higher education, living in London and currently concurrently smoking and vaping were associated with ever having used HTPs or nicotine pouches; men were also more likely to have ever used nicotine pouches. Curiosity was the most common reason for use or trial of HTPs and a majority of those who had ever used them found them at least as satisfying as smoking. Nearly half of those not currently using HTPs expressed interest in using them: interest was lower among those aged over 65 than in all other age groups and among those not smoking or vaping compared with those smoking only but higher among those smoking and vaping.

Awareness of HTPs appears to have increased in Great Britain since 2017, when awareness was between 10% and 18% of current or recent past smokers/vapers [24], although differences between surveys and measures make comparisons difficult. Overall, current use appears not to have increased (for example 2.0% among current smokers in 2017 compared with 1.7% among exclusive smokers in this survey). Prevalence among smokers, recent ex-smokers and vapers in the present survey was consistent with previous surveys [29].

Despite low awareness, current use of nicotine pouches was similar to current use of HTP in this sample. In comparison to other countries, such as Japan and South Korea [27,28], use of HTPs is much lower, although this might also reflect reduced accessibility of vaping products in those countries compared with Great Britain. It is also of note that not all ever users of nicotine pouches reported having seen them for sale, suggesting trial of other people’s products or free samples given out as marketing strategy. There are no prior surveys in Great Britain or other countries of nicotine pouch use for comparison.

The characteristics of those with some experience of using HTPs and nicotine pouches were remarkably consistent, with those under 45, those with at least some university experience and those who were both smoking and vaping being more likely to have some experience. The socio-economic differences for HTPs may reflect the retail cost of HTPs, which for a starter pack is approximately £40 (approximately $ 54 as of December 2020). However, cheaper cost was a reason for using that current HTP users were more likely to state than ever users. This reflects that ongoing costs are lower as once the HTP device has been purchased, a pack of tobacco sticks are approximately half the price of a pack of cigarettes. The price of a tin of 20 nicotine pouches is approximately equivalent to a pack of 20 cigarettes, so the socio-economic differences may instead reflect differential accessibility/marketing to different socio-economic groups or areas, or that smokers are looking for cheaper substitutes for smoking overall.

That people who both smoked and vaped, but not those exclusively smoking or vaping, were more likely to have tried HTPs and pouches may reflect their desire to find a product to help them stop smoking given vaping has not resulted in their stopping smoking. Similarly, smokers and vapers who were not HTP users were more likely to be interested in trying HTP compared with exclusive smokers. Current HTP users were indeed more likely to choose quitting smoking as a reason for use compared to HTP triers. However, nearly all the reasons supplied in the questionnaire were more likely to be endorsed by current users compared to triers, and several other reasons were more popular than quitting smoking for current users.

Overall, that nearly 90% of current HTP users found HTPs at least as satisfying as cigarettes suggests that HTPs could be a successful substitute for cigarettes in those using them to quit smoking. Fewer triers (45%) found HTPs as satisfying as cigarettes suggesting that their impact differs across users.

To our knowledge, this is the first study to provide information about use of nicotine pouches and detailed evidence on HTP use and interest, but results need to be viewed in the light of some limitations. Results provide a snapshot of the situation at the time in what is a constantly moving field. Data were from a survey with respondents self-reporting all measures and no verification of responses was possible. Particularly as the products of interest are relatively new, confusion may have occurred with other products such as vaping products or chewing tobacco pouches. The survey did, however, include explanations and pictures at the beginning of the survey and these could be accessed throughout the survey to reduce any confusion. Despite an overall large sample, the low prevalence of use meant we were only able to assess associations with ever use, not the more relevant current use and sample sizes were small for some analyses. Middle aged adults were over-represented, providing less information particularly about older adults. The sample was restricted to adults with experience of nicotine use, so cannot provide information about uptake or interest in these products among those who have never used nicotine products or smoked.

To assess public health impact, future research should therefore assess interest and use among nicotine-naïve groups as well as the effects of use of HTPs or nicotine pouches on smoking cessation or reduction. If these products help smokers to stop and/or are significantly less harmful, then more needs to be done to increase their use among exclusive smokers. Regulatory restrictions may then need to be reconsidered, although a cross-country study only found small differences between countries varying even on legal availability [29].

## 5. Conclusions

In this sample of adults with a history of nicotine use, very few currently used heated tobacco products or nicotine pouches. Satisfaction with and interest in HTPs were substantial. The low level of use is unlikely to substantially reduce the public health impact of smoking.

## Figures and Tables

**Table 1 ijerph-18-08852-t001:** Sample description, *n* = 3883.

Socio-Demographics	%	*n*	Nicotine Product Use	%	*n*
Gender			Smoking		
Male	50.3	1955	Daily cigarettes	48.0	1864
Female	49.7	1928	Non-daily cigarettes	15.7	610
Age			Other tobacco	5.6	219
18–24	8.7	339	Stopped smoking in last year	9.8	380
25–34	18.1	703	Stopped smoking over a year ago	19.1	742
35–44	21.2	823	Never smoked	1.8	68
45–54	22.5	872	Vaping		
55–65	17.0	659	Daily	24.4	949
Over 65	12.5	487	Non-daily	12.8	498
Ethnicity			Tried a few times	23.1	897
UK white	84.2	3269	Stopped vaping in last year	5.2	201
Other white	6.6	256	Stopped vaping over a year ago	5.8	226
Mixed	2.6	102	Never vaped	28.6	1112
Asian	4.0	154	Combined smoking and vaping		
Black	1.5	59	Smoking only	46.9	1822
Other, not reported	1.1	43	Smoking and vaping	22.4	871
Education			Vaping only	14.8	576
No university/ not reported	54.1	2101	Not smoking or vaping	15.8	614
Some university	45.9	1782	Heated Tobacco		
UK region			Ever heard	34.8	1353
England excluding London	69.1	2683	Seen for sale	10.1	392
Greater London	15.3	596	Ever used	6.2	242
Wales	4.5	176	Current frequency of use ^1^		
Scotland	8.9	344	Daily ^2^	18.6	45
Northern Ireland	2.2	84	Less than daily, at least once a week ^2^	14.5	35
			Less than weekly, at least once a month ^2^	18.2	44
			Less than monthly	12.4	30
			Tried a few times	16.1	39
			Tried once	16.5	40
			Don’t know	3.7	9
			Nicotine pouches		
			Ever heard	15.9	616
			Seen for sale	3.1	122
			Ever used	4.4	171
			Current frequency of use ^1^		
			Daily ^2^	20.5	35
			Less than daily, at least once a week ^2^	19.9	34
			Less than weekly, at least once a month ^2^	19.9	34
			Less than monthly	9.9	17
			Tried a few times	11.7	20
			Tried once	18.1	31
			Don’t know	0	0

^1^ Among ever users. ^2^ Categorised as current use.

**Table 2 ijerph-18-08852-t002:** Prevalence of current and ever use (includes current use) of heated tobacco products and nicotine pouches, associations between ever use and respondent characteristics and prevalence of and associations with interest in trying HTPs.

	Heated Tobacco Products							Nicotine Pouches			
	Prevalence, *n* = 3883		Adjusted Associations for Ever Use ^1^		Interest, *n* = 1199	Adjusted Associations for Interest		Prevalence, *n* = 3883		Adjusted Associations for Ever Use ^1^	
	Current	Ever	OR (95 CI)	*p*		OR (95 CI)	*p*	Current	Ever	OR (95 CI)	*p*
Gender											
Male	3.5	6.2	1.17 (0.89–1.55)	0.26	46.8	0.98 (0.76–1.27)	0.89	2.9	5.1	1.79 (1.29–2.49)	0.001
Female	2.9	6.2	Ref		50.8	Ref		2.4	3.7	Ref	
Age											
18–24	6.5	10.9	5.89 (2.55–13.58)	<0.001	59.1	3.60 (1.99–6.53)	<0.001	5.3	7.4	7.53 (2.55–22.24)	<0.001
25–34	6.7	11.2	5.36 (2.41–11.92)	<0.001	64.5	4.26 (2.58–7.04)	<0.001	6.1	9.0	8.04 (2.85–22.69)	<0.001
35–44	4.4	8.4	4.60 (2.07–10.2)	<0.001	58.0	3.70 (2.30–5.96)	<0.001	3.9	6.2	6.09 (2.16–17.15)	0.001
45–54	0.9	3.2	2.05 (0.88–4.75)	0.096	41.7	2.00 (1.28–3.12)	0.002	0.6	2.1	2.33 (0.78–6.96)	0.131
55–65	1.4	3.3	2.06 (0.87–4.89)	0.101	42.6	1.90 (1.21–2.99)	0.006	0.8	1.5	1.54 (0.48–4.97)	0.468
Over 65	0.4	1.4	Ref		27.7	Ref		5.3	0.8	Ref	
Ethnicity											
Not white	5.6	10.3	0.97 (0.65–1.43)	0.871	53.6	0.86 (0.56–1.32)	0.49	5.6	8.1	1.03 (0.66–1.59)	0.908
White	3.0	5.8	Ref		47.9	Ref		2.4	4.0	Ref	
Education											
Some university	5.3	9.6	2.11 (1.56–2.84)	<0.001	53.9	1.13 (0.88–1.44)	0.34	4.7	7.0	2.09 (1.45–3.00)	<0.001
No university	1.4	3.4	Ref		43.4	Ref		1.0	2.2	Ref	
Region											
Greater London	7.9	11.9	1.75 (1.28–2.40)	<0.001	54.4	1.28 (0.92–1.77)	0.140	7.0	9.2	2.10 (1.45–3.03)	<0.001
Wales	0	2.3	0.42 (0.15–1.16)	0.094	48.1	0.95 (0.53–1.72)	0.87	0.6	1.1	0.34 (0.08–1.39)	0.132
Scotland	2.0	3.8	0.66 (0.37–1.19)	0.169	38.9	0.82 (0.51–1.30)	0.40	2.6	4.9	1.54 (0.89–2.66)	0.122
Northern Ireland	1.2	3.6	0.64 (0.20–2.07)	0.451	56.7	2.19 (0.96–5.02)	0.063	7.0	3.6	1.13 (0.34–3.71)	0.845
England excl London	2.6	5.6	Ref		47.6	Ref		1.9	3.5	Ref	
Smoking /Vaping											
Not smoking or vaping	1.6	3.4	0.67 (0.41–1.09)	0.107	23.3	0.29 (0.19–0.44)	<0.001	2.0	3.4	1.05 (0.62–1.77)	0.86
Smoking and vaping	8.6	13.2	2.26 (1.67–3.07)	<0.001	62.6	1.39 (1.02–1.89)	0.038	6.8	9.5	2.25 (1.56–3.25)	<0.001
Vaping only	1.6	3.5	0.76 (0.46–1.26)	0.29	42.9	0.77 (0.55–1.06)	0.109	0.9	2.1	0.72 (0.38-1.36)	0.31
Smoking only	1.7	4.7	Ref		49.9	Ref		1.5	3.0	Ref	

^1^ Adjusted for gender, age, education, ethnicity, region and smoking and vaping status.

**Table 3 ijerph-18-08852-t003:** Reasons for use of HTPs and satisfaction relative to smoking, *n* = 242 ever users.

	Agreement, %			Comparison Current vs. Tried	
	Overall	Current Users	Tried	χ^2^	*p*
Which of the following were reasons for your using heat-not-burn products?	*n* = 242	*n* = 124	*n* = 109		
Just to give it a try/ I was curious about them	79.8	75.0	85.3	6.78	0.028
They smell better than cigarettes	70.8	76.6	64.2	6.13	0.046
No tobacco smoke	69.4	67.9	68.7	0.37	0.89
They don’t produce ash or butts	66.5	62.9	70.0	5.31	0.071
They may be more socially acceptable	66.1	74.2	56.9	7.82	0.019
They make it easier for you to cut down on the number of cigarettes you smoke	65.2	75.8	53.2	13.55	0.001
I like the flavours	64.4	81.5	45.0	34.32	<0.001
They may not be as bad for your health as smoking tobacco cigarettes	63.9	66.9	60.6	11.8	0.002
The technology	63.5	71.8	54.1	8.14	0.015
They might help you quit smoking	61.8	66.9	56.0	8.11	0.016
Because I enjoy it	60.1	71.0	47.7	15.06	<0.001
So you can use them in places where smoking tobacco or cigarettes is banned	59.2	66.1	51.4	8.08	0.018
Because I like the taste	58.8	75.0	40.4	29.11	<0.001
Friends or family use them	58.4	67.7	47.7	10.64	0.004
They are cheaper than cigarettes	54.5	65.3	42.2	16.69	<0.001
A health professional advised you to do so	43.3	62.9	21.2	42.28	<0.001
Other	30.5	42.7	16.5	30.52	<0.001
How satisfying is using a heat-not-burn product compared to smoking tobacco cigarettes? ^1^					
More satisfying	29.0	37.1	14.5		
Equally satisfying	43.0	50.0	30.4	40.15	<0.001
Less satisfying	25.4	12.1	49.3		
Don’t know	2.6	0.8	5.8		

^1^*n* = 40 who tried once excluded.

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
