# Peer review of "Heated Tobacco Products and Nicotine Pouches: A Survey of People with Experience of Smoking and/or Vaping in the UK"

_ijerph, 2021, doi:10.3390/ijerph18168852_

Round 1

Reviewer 1 Report

“Heated tobacco products and nicotine pouches: A survey of people with
experience of smoking and/or vaping in the UK”

Comments on the article by L.S. Brose et al submitted to the
International Journal of Environmental Research and Public Health (MS 1295968)

_______________________________________________________________

                                                      Author   :    P.N. Lee

                                                      Date       :    5th July 2021

The authors describe results from a survey conducted in the UK.  While the results are clearly presented, I felt the final sentence in the abstract “If associations with age and education persist, the products will not help reduce smoking-related inequalities” puts over a very misleading picture, tending to imply that the introduction of heated tobacco products (HTPs) and nicotine pouches (NPs) is of little value.

In the first place, the paper does not adequately clarify clearly enough the substantial reductions in risk of smoking-related disease if a smoker switches to these products.  For IQOS, one of the HTPs, there is mention that the FDA concluded that switching to them “reduced exposure to harmful or potentially harmful chemicals”.  However, there is no mention of the fact that in Japan, where sales of IQOS have been substantial, sales of cigarettes have declined substantially.  More seriously, for NPs there is no mention of their similarity to snus, and that substantial epidemiological evidence clearly demonstrates that any harmful effects from their use are very much less than those for smoking.

The conclusion about not helping reduce smoking-related inequalities is in any case not well-justified, and may well be incorrect.  Consider a hypothetical simple example in which 30% of higher social class people smoke, while 50% of lower social class people do.  Assuming that smoking doubles risk of mortality then compared to the risk in non-smokers, that in the lower class is 1.5, while that in the higher class is 1.3, an “inequality” ratio of 1.5/1.3 = 1.154, or a difference of 1.5-1.3 = 20% of the non-smoker rate. 

Now suppose that nicotine pouches become popular and that they have a relative risk of 1.2.  If 80% of upper class individuals switch, their overall risk is 1.108, while if only 60% of lower social class do, their risk is 1.260, a slightly smaller ratio of 1.137, and a smaller difference of 15.2%.

Clearly there are many assumptions involved, but if the final sentence is to stay (which it probably should not) it should be very clearly justified.  It would be better to end with a sentence hoping that more smokers switch to these products, which seems to me to be much more relevant than any statement about social class differences.

Otherwise I only had some minor points.

  1. Conclusion in abstract, penultimate sentence. Split into two sentences after “3%”.  At present the sentence has two main verbs.
  2. Introduction, paragraph 2, sentence 3. Amend “all produced” to “each produced”.  No company produces all of them.
  3. There is mention in the methods section of “bivariate and multivariable” logistic regression, and in Table 2 of adjusted associations, but it is not made clear what variables are being adjusted for, and whether the results come from bivariate or multivariable adjustment.

Author Response

We thank the reviewers for their insightful comments and suggestions. A response is provided below each comment. We feel these edits have improved our manuscript and hope it is now acceptable for publication in IJERPH.  

Reviewer 1:

The authors describe results from a survey conducted in the UK.  While the results are clearly presented, I felt the final sentence in the abstract “If associations with age and education persist, the products will not help reduce smoking-related inequalities” puts over a very misleading picture, tending to imply that the introduction of heated tobacco products (HTPs) and nicotine pouches (NPs) is of little value. 

Response: Please see our response at the end of the reviewer’s major comments.  

In the first place, the paper does not adequately clarify clearly enough the substantial reductions in risk of smoking-related disease if a smoker switches to these products.  For IQOS, one of the HTPs, there is mention that the FDA concluded that switching to them “reduced exposure to harmful or potentially harmful chemicals”.  However, there is no mention of the fact that in Japan, where sales of IQOS have been substantial, sales of cigarettes have declined substantially.  More seriously, for NPs there is no mention of their similarity to snus, and that substantial epidemiological evidence clearly demonstrates that any harmful effects from their use are very much less than those for smoking. 

The conclusion about not helping reduce smoking-related inequalities is in any case not well-justified, and may well be incorrect.  Consider a hypothetical simple example in which 30% of higher social class people smoke, while 50% of lower social class people do.  Assuming that smoking doubles risk of mortality then compared to the risk in non-smokers, that in the lower class is 1.5, while that in the higher class is 1.3, an “inequality” ratio of 1.5/1.3 = 1.154, or a difference of 1.5-1.3 = 20% of the non-smoker rate.  

Now suppose that nicotine pouches become popular and that they have a relative risk of 1.2.  If 80% of upper class individuals switch, their overall risk is 1.108, while if only 60% of lower social class do, their risk is 1.260, a slightly smaller ratio of 1.137, and a smaller difference of 15.2%. 

Clearly there are many assumptions involved, but if the final sentence is to stay (which it probably should not) it should be very clearly justified.  It would be better to end with a sentence hoping that more smokers switch to these products, which seems to me to be much more relevant than any statement about social class differences. 

Response: Thank you for these points. We have changed the abstract conclusion: “In this sample of adults with a history of nicotine use, very few currently used heated tobacco products and nicotine pouches. Satisfaction with and interest in HTPs were substantial. The low level of use is unlikely to substantially reduce the public health impact of smoking.” 

We have also edited the conclusion in the main body to be the same as in the abstract.  

We have added information on HTP in Japan and on snus to the introduction.: “Notably, in Japan, the introduction and rapid increase in use of HTP (Hori et al., 2020; Odani and Tabuchi, 2021) has led to an acceleration in the decline of cigarette sales (Cummings et al., 2020; Stoklosa et al., 2020).”  

“Nicotine pouches entered the UK market in 2018-19. In appearance, they resemble snus, a type of tobacco pouch widely used in Sweden with substantially reduced harm compared with smoking (Lee, 2013; Hatsukami and Carroll, 2020). However, in contrast to snus, nicotine pouches do not contain tobacco but nicotine-containing powder (Robichaud et al., 2019).” 

Otherwise I only had some minor points. 

  1. Conclusion in abstract, penultimate sentence. Split into two sentences after “3%”.  At present the sentence has two main verbs. 

Response: We have split the sentence into two. 

  1. Introduction, paragraph 2, sentence 3. Amend “all produced” to “each produced”.  No company produces all of them. 

Response: Amended as suggested.  

  1. There is mention in the methods section of “bivariate and multivariable” logistic regression, and in Table 2 of adjusted associations, but it is not made clear what variables are being adjusted for, and whether the results come from bivariate or multivariable adjustment. 

Response: We have added the missing information to the analysis section: “. Multivariable logistic regressions included gender, age, education, ethnicity, region and smoking and vaping status.” And “The results section reports multivariable (adjusted) analysis unless otherwise reported.” Information has also been added as a footnote to Table 2: “Adjusted for gender, age, education, ethnicity, region and smoking and vaping status.” 

Reviewer 2 Report

This paper is a useful addition to the understanding of consumer's knowledge and attitudes concerning heated tobacco products and non-tobacco nicotine pouches, relatively recent newcomers to the range of safer nicotine products. There are no major issues with the way the study was conducted and reported, and it fits with the focus of  the special issue.

The only problem is that the study was carried out in 2019, and in a fast moving market for new nicotine products in needs to be historically contextualised. (And if I may say, why do academics take so long to get policy relevant information into the public domain!).

The authors found higher awareness of HTP than nicotine pouches. I have seen more recent data for the UK which suggests the opposite, with much higher awareness of pouches than HTP. This might be connected with (a) pouches coming onto the market later than HTP - which might explain the lower awareness of them in 2019 and (b) the different regulations for the promotion/advertising of HTP versus pouches. The authors should add information about the dates that HTP and pouches came onto the market, and caution the reader that their work is an historical snapshot.

The comment about FDA and IQOS (first para, page 2) re: reduced exposure and reduced risk could do with some small clarification about the FDA decission. In 2016 PMI applied for both “risk modification” and “exposure modification” orders.

There are two types of MRTP orders: to quote the FDA press release "the FDA may issue: a “risk modification” order or an “exposure modification” order. The company had  requested both types of orders for the IQOS Tobacco Heating System. After reviewing the available scientific evidence, public comments and recommendations from the Tobacco Products Scientific Advisory Committee, the FDA determined that the evidence did not support issuing risk modification orders at this time but that it did support issuing exposure modification orders for these products. This determination included a finding that issuance of the exposure modifications orders is expected to benefit the health of the population as a whole. FDA’s authorization for exposure modification is based on current evidence on exposure in the absence of long term studies and states that such an order can be made if “[t]he scientific evidence that is available without conducting long‐term epidemiological studies demonstrates that a measurable and substantial reduction in morbidity or mortality among individual tobacco users is reasonably likely in subsequent studies (section 911(g)(2)(A) of the FD&C Act).”

Author Response

We thank the reviewer for their insightful comments and suggestions. A response is provided below each comment. We feel these edits have improved our manuscript and hope it is now acceptable for publication in IJERPH.  

Reviewer 2 

This paper is a useful addition to the understanding of consumer's knowledge and attitudes concerning heated tobacco products and non-tobacco nicotine pouches, relatively recent newcomers to the range of safer nicotine products. There are no major issues with the way the study was conducted and reported, and it fits with the focus of  the special issue. 

Response: Thank you for the positive evaluation. 

The only problem is that the study was carried out in 2019, and in a fast moving market for new nicotine products in needs to be historically contextualised. (And if I may say, why do academics take so long to get policy relevant information into the public domain!). 

Response: Academics usually need to fit in research in between education and administration. Leading education during a global pandemic also came with substantial challenges, leaving little time for research. 

The authors found higher awareness of HTP than nicotine pouches. I have seen more recent data for the UK which suggests the opposite, with much higher awareness of pouches than HTP. This might be connected with (a) pouches coming onto the market later than HTP - which might explain the lower awareness of them in 2019 and (b) the different regulations for the promotion/advertising of HTP versus pouches. The authors should add information about the dates that HTP and pouches came onto the market, and caution the reader that their work is an historical snapshot. 

Response: We have added information to the introduction: “IQOS (Philip Morris International), the first HTP to become available in the UK from late 2016 […]” and later “Nicotine pouches entered the UK market in 2018-19. 

We have added to the limitations in the discussion: “Results provide a snapshot of the situation at the time in what is a constantly moving field. “ 

The comment about FDA and IQOS (first para, page 2) re: reduced exposure and reduced risk could do with some small clarification about the FDA decission. In 2016 PMI applied for both “risk modification” and “exposure modification” orders. 

There are two types of MRTP orders: to quote the FDA press release "the FDA may issue: a “risk modification” order or an “exposure modification” order. The company had  requested both types of orders for the IQOS Tobacco Heating System. After reviewing the available scientific evidence, public comments and recommendations from the Tobacco Products Scientific Advisory Committee, the FDA determined that the evidence did not support issuing risk modification orders at this time but that it did support issuing exposure modification orders for these products. This determination included a finding that issuance of the exposure modifications orders is expected to benefit the health of the population as a whole. FDA’s authorization for exposure modification is based on current evidence on exposure in the absence of long term studies and states that such an order can be made if “[t]he scientific evidence that is available without conducting long‐term epidemiological studies demonstrates that a measurable and substantial reduction in morbidity or mortality among individual tobacco users is reasonably likely in subsequent studies (section 911(g)(2)(A) of the FD&C Act).” 

Response: We have clarified the statement about the FDA decision:  

"Philip Morris International had requested a risk modification and an exposure modification order for IQOS in the US. The US Food and Drug Administration (FDA) authorised IQOS to be marketed as a modified risk tobacco product (US Food and Drug Administration, 2020). The FDA concluded that, based on current evidence on exposure in the absence of long-term studies, switching completely from smoking to IQOS reduced exposure to harmful or potentially harmful chemicals. However, they did not approve that IQOS use reduces the risk of tobacco-related diseases or reduced harm relative to continued cigarette smoking. "

Reviewer 3 Report

Materials and methods section: Please, include a subsection of statistical analysis and explain the analysis performed in Tables 2 and 3.

You need a discussion section and need a conclusion according to you the results obtained.

Author Response

We thank the reviewer for their insightful comments and suggestions. A response is provided below each comment. We feel these edits have improved our manuscript and hope it is now acceptable for publication in IJERPH.  

Reviewer 3 

Materials and methods section: Please, include a subsection of statistical analysis and explain the analysis performed in Tables 2 and 3. 

Response: We have split the subsection of analysis and sample into two and have extended the previously incomplete description of the analysis which also includes the analyses performed in Tables 2 and 3. The sections in full now read:  

“2.2. Sample 

All 3,883 respondents were included to address aims 1 and 2. To address aim 3, the 242 respondents who had ever used HTPs were included for reasons for use and the subset of 193 respondents who had used HTPs more than once to describe satisfaction. To address aim 4, the 1,199 respondents who had not used HTPs or had tried them up to a few times only were included in analysis. 

2.3. Analyses 

To address aim 1, the prevalence of having heard of, ever use and at least monthly us (current use) of HTP and nicotine pouches was described.  

To address aim 2, associations between ever use of HTPs and nicotine pouches and socio-demographics and smoking and vaping status were assessed using bivariate and multivariable logistic regressions. Multivariable logistic regressions included gender, age, education, ethnicity, region and smoking and vaping status. Only ever use was used in regressions because current use was low. The results section reports multivariable (adjusted) analysis unless otherwise reported. 

To address aim 3, frequency of responses to a list of possible reasons for use were described and compared using chi-square statistics for current users of HTPs and for those who had only tried HTPs or were using it less than monthly. Satisfaction with HTPs compared with cigarettes was described for those two groups.  

To address aim 4, proportions interested in trying HTPs were described; bi-variate and multivariable logistic regressions were used to assess associations with socio-demographics and smoking and vaping status. Multivariable logistic regressions included gender, age, education, ethnicity, region and smoking and vaping status and adjusted results are reported unless otherwise reported. In-formation on reasons for use, satisfaction or interest in use was not available for nicotine pouches.”  

You need a discussion section and need a conclusion according to you the results obtained. 

Response: We have changed the heading and added a separate conclusion section: “In this sample of adults with a history of nicotine use, very few currently used heated tobacco products or nicotine pouches. Satisfaction with and interest in HTPs were substantial. The low level of use is unlikely to substantially reduce the public health impact of smoking.” 

This manuscript is a resubmission of an earlier submission. The following is a list of the peer review reports and author responses from that submission.